# The Incidence, Survival, and HPV Impact of Second Primary Cancer following Primary Oropharyngeal Squamous Cell Carcinoma: A 20-Year Retrospective and Population-Based Study

**DOI:** 10.3390/v15010034

**Published:** 2022-12-22

**Authors:** Lasse Andersen, Kathrine Kronberg Jakobsen, Amanda-Louise Fenger Carlander, Martin Garset-Zamani, Jeppe Friborg, Katalin Kiss, Rasmus L. Marvig, Caroline Olsen, Finn Cilius Nielsen, Elo Andersen, Christian Grønhøj, Christian von Buchwald

**Affiliations:** 1Department of Otorhinolaryngology, Head and Neck Surgery and Audiology, Rigshospitalet, University of Copenhagen, DK-2100 Copenhagen, Denmark; 2Department of Oncology, Herlev Hospital, University of Copenhagen, DK-2730 Herlev, Denmark; 3Department of Pathology, Rigshospitalet, University of Copenhagen, DK-2100 Copenhagen, Denmark; 4Center for Genomic Medicine, Rigshospitalet, University of Copenhagen, DK-2100 Copenhagen, Denmark

**Keywords:** second primary cancer, oropharyngeal cancer, human papillomavirus, squamous cell carcinoma

## Abstract

Second primary cancer (SPC) is the second most common cause of death among patients diagnosed with head and neck cancer. This study examined the risk of SPC following oropharyngeal squamous cell carcinoma (OPSCC) and the impact of human papillomavirus (HPV) on survival following SPC. The study was a population-based, retrospective study including all patients diagnosed with OPSCC in eastern Denmark from 2000–2020 who received curative intended treatment. The incidence rate ratio (IRR), age-adjusted incidence rates (AAIR), and hazard ratios (HR) were calculated. A total of 2584 patients with primary OPSCC were included (median follow-up time: 3.1 years), with 317 patients (12.3%) diagnosed with SPC. The risk of SPC was approximately five times the occurrence of cancer in the general population (IRR: 4.96). The median time to SPC after a primary OPSCC was 2.0 years (interquartile range (IQR) = 0.6–4.2 years). HPV-positive (HPV+) patients had a significantly longer median time to SPC, and a significant better survival compared to HPV-negative (HPV-) patients. SPC was most frequently found in lungs, head, and neck (LHN) for HPV- OPSCC patients and lungs followed by gender-specific (prostate, ovaries, or endometrium) for HPV+ OPSCC. There was a significant difference between the two groups when distributed between “within” or “outside” LHN. Patients with SPC outside LHN had a significant better overall survival. This knowledge should be considered during post-treatment surveillance and might guide targeted imaging.

## 1. Introduction

The global incidence of oropharyngeal squamous cell carcinoma (OPSCC) is estimated to be approximately 93,000 new patients per year [1]. During the past decades, the incidence of OPSCC has been increasing in the Western world [2], mainly due to an increase in human papillomavirus-positive (HPV+) OPSCCs [3,4,5]. We previously reported a high and increasing incidence of OPSCC with age-adjusted incidence rates (AAIR) per 100,000 from 1.8 in 2000 to 5.1 in 2017 in eastern Denmark, primarily driven by an increase in HPV+ OPSCC with 65% of OPSCCs being HPV+, with HPV16 as the predominant genotype (86%) [4,6,7,8,9]. In addition to HPV, well-known risk factors for developing OPSCC are tobacco smoking and alcohol consumption [10].

HPV+ OPSCC patients have a better prognosis and demonstrate distinct clinical, histopathological, and genetic characteristics compared to HPV-negative (HPV-) OPSCC patients [10,11,12,13,14]. The tumour suppressor p16 is often overexpressed in HPV+ OPSCCs, and may therefore be used as a surrogate marker for HPV DNA. Solely double positivity for p16 and HPV DNA (p16+/HPV DNA+) implicates a better prognosis [15].

Secondary primary cancer (SPC) is the second most common cause of death among patients with head and neck squamous cell carcinomas (HNSCCs), including OPSCC [16,17,18]. SPCs are divided into two groups: synchronous SPCs diagnosed at the same time or within six months after the primary cancer and metachronous SPCs diagnosed more than six months after [19]. The majority of SPCs are metachronous, while synchronous cancer is a relatively rare event but is clinically important since it might affect treatment regimen [20].

Alcohol consumption and tobacco smoking are notable risk factors for developing SPC, especially regarding cancers localized in the upper aerodigestive tract, lungs, and oesophagus [21]. Studies have shown that p16+ OPSCCs have lower risk of SPC [22], primarily driven by a decrease in SPCs within the upper aerodigestive tract [23,24].

It is important to better characterize the risk of SPC in OPSCC and the relation to HPV status to improve screening, treatment, and follow-up strategies and hence oncologic outcomes.

To our knowledge, the incidence and pattern of SPC in patients with OPSCC have not been evaluated in a population-based study in a high-HPV-prevalence area in patients with known HPV DNA and p16 status. The aim of this study was to examine the risk of SPC in patients with OPSCC in eastern Denmark from 2000–2020. Further, we evaluated the survival following SPC in relation to HPV status.

## 2. Materials and Methods

### 2.1. Patient Cohort

The cohort consisted of all patients diagnosed with OPSCC between 2000–2020 in eastern Denmark, comprising approximately 46% of the Danish population. All patients were diagnosed and treated in head and neck surgery departments and/or oncology departments at public university hospitals in Eastern Denmark. All patients registered with OPSCC in the Danish Head and Neck Cancer Group (DAHANCA) database and the Danish Pathology Databank and identified by a personal identification number were included. All data were gathered in a REDcap^®^ [25,26] database containing information on both clinical and tumour characteristics at the time of OPSCC diagnosis (baseline), including age, sex, sublocation of OPSCC (palatine tonsils, base of tongue, and ‘other sublocations’ (uvula, soft palate, oropharyngeal wall)), Union for International Cancer Control 8th edition (UICC8) TNM-classification, tumour stage, tobacco-smoking status (never smoker, former smoker, current smoker), alcohol consumption (no abuse, previous abuse, or current abuse, defined by >7 units per week for females and >14 units per week for men), histopathological type (non-keratinizing, keratinizing/hybrid/other), HPV DNA status, p16 status, WHO performance score, follow-up time, prior cancers, chemotherapy type, and SPC. 

### 2.2. HPV Analysis 

Histological specimens were reviewed for p16 staining by immunohistochemistry (IHC) by specialized pathologists and analysed for HPV DNA by polymerase chain reaction (PCR) using general primers GP5+/6+ as previously described [5,6,15]. HPV+ patients were defined as being positive if they were positive for both HPV DNA and p16 [5,6].

### 2.3. SPC Definition

SPC was defined as a new malignancy fulfilling the following criteria:Basal cell carcinoma of the skin was excluded;The tumour had to be biopsy verified and identified through The Danish Pathology Databank;Cancers in the oropharynx were considered a recurrence within the first five years after primary diagnosis. OPSCC later than five years after primary diagnosis was considered a SPC unless specifically stated as a recurrence in medical journals;Cancers in the head and neck area that were evaluated to be a different localization than the oropharynx by a multidisciplinary team or evaluated to be histologically or cytologically different were considered a SPC unless specifically stated as a recurrence in medical files;Synchronous and metachronous cancers were pooled, and therefore, no time limit was used.

### 2.4. Statistical Analysis

Statistical analysis was performed in R statistics version 4.1.3 [27]. Linearity of the continuous covariables was tested by plotting the Martingale residuals against continuous covariables (see Appendix A). We further tested for proportionality of the variables by log-minus-log curves and Schoenfeld residual plots (see Appendix A).

Continuous variables were reported as median values with range or interquartile range (IQR) and categorical variables as frequencies. To test the variables for significance, we used Pearson’s chi-square test for the binomial categoric covariables and a *t*-test for the quantitative covariables distributed in two groups. On M-classification, we used Fischer’s exact test due to the low number of cases. We considered a *p*-value <0.05 as statistically significant.

Univariable and multivariable Cox regression analysis were performed in R with the packages Survminer and Survival [27]. In the multivariable Cox regression analysis, we adjusted for the variables age, sex, sublocation for the primary OPSCC, UICC8 stage, UICC8 T-classification, UICC8 N-classification, smoking status, and HPV status. UICC8 M-classification was excluded due to the low occurrence to prevent statistical instability. Variables were selected based on clinical experience as well as prior studies. The time variable was the time from the date of OPSCC diagnosis to either the date of SPC, time to last follow-up, or death. An incidence rate ratio (IRR) was calculated for all included cases with the general population in Denmark used as reference. Data were retrieved from the national health register administrated by the Danish Health Data Authority and from the Statbank administered by Statistics Denmark [28]. 

Age-adjusted incidence rates (AAIR) per 100,000 were calculated using the direct method with the EpiTools [27] package using the eastern Denmark population and WHO world standard population as reference [29]. 

Kaplan–Meier curves were made to compare survival for patients with SPC stratified by HPV status and SPC localization with baseline at the date of biopsy-verified SPC diagnosis. Significance was assessed with the log-rank method. The one-, three-, and five-year overall survival was calculated for both HPV- and HPV+ patients.

## 3. Results

### 3.1. Characteristics of the Cohort at the Time of Diagnosis

Among a total of 2854 patients diagnosed with OPSCC, 264 patients were excluded due to non-curative treatment intent. We further excluded six patients with synchronous bilateral tonsillar squamous cell carcinoma (BiTSCC) since the mechanism is poorly understood and therefore is controversial [19]. In total, we enrolled 2584 patients with biopsy-verified OPSCCs. Most patients received radiotherapy either combined with surgery, chemotherapy, or both surgery and chemotherapy (*n* = 2370, 91.7%). Cisplatin was the predominant drug used for chemotherapy (*n* = 1094, 91.6%). For more baseline characteristics, see Table 1**.**

### 3.2. Factors Associated with the Risk of SPC

A total of 317 (12.3%) patients were diagnosed with a SPC following their OPSCC. Median time to SPC was 2.0 years (IQR = 0.6–4.2 years). HPV+ OPSCC had a significantly longer median time to SPC (*p* = 0.006) compared to HPV- OPSCC with a median time to SPC of 2.8 years (IQR = 0.7–5.2 years) for HPV+ OPSCC patients and 1.7 years (IQR = 0.5–3.3 years) for HPV- OPSCC patients, respectively. Patients having a SPC were significantly older (median age 64 years (37–86 years) versus 61 years (30–92 years), *p* < 0.001), were more likely to have lymph node metastasis (*p* < 0.001), and showed a significant difference in UICC8 N-classification. Overall, the two groups were similar in terms of sex, UICC8 T- and M-classification, tumour stage, performance score, chemotherapy type, and prior cancer. See Table 1.

The multivariable analysis showed an increased hazard ratio (HR) for age (HR = 1.04, *p* < 0.001) and UICC8 stage II (HR = 1.80, *p* = 0.004) and trended toward increased HR in UICC8 stage III (HR = 1.60, *p* = 0.057) and UICC8 stage IV (HR = 1.94, *p* = 0.058). HPV+ patients showed a significantly decreased HR (HR = 0.56, *p* = 0.001) as well as N2 cases (HR = 0.61, *p* = 0.036). No association in the risk of SPC was found between sex (*p* = 0.287) or sublocations (base of tongue *p* = 0.123 and “other” *p* = 0.887), with palatine tonsils as reference. Further, UICC8 T2 (*p* = 0.302), T3 (*p* = 0.072), and T4 (*p* = 0.722); UICC8 N-classification N1 (*p* = 0.129) and N3 (*p* = 0.466); AND smoking status (“former smoker”, *p* = 0.112 and “current smoker”, *p* = 0.594) were not significant. See Table 2. 

### 3.3. Incidence Trends and Characteristics of SPC

We observed an overall increasing trend in AAIR of SPC per 100,000 from 2000–2020, which is shown in Figure 1. The risk of SPC following OPSCC was almost five times greater than the occurrence of cancer in the general population in Denmark with an IRR of 4.96 [95%CI: 4.33–5.68]. 

The localizations of SPC were most frequently the lungs (30.0%), head and neck (16.4%), and the gastrointestinal tract (14.8%). Stratified by HPV status, we found that the lungs (35.6%), the head and neck (19.4%), and the gastrointestinal tract (18.1%) were also the most predominant localizations in patient diagnosed with HPV- OPSCC. For HPV+ OPSCC patients, the most frequent SPC localization was the lungs (24.2%), followed by gender-specific localization (prostate, ovaries, or endometrium) (14.6%) and the head and neck (13.4%). There was a significant difference between the two groups when distributed between “within” or “outside” lungs, head, and neck (LHN) (*p* = 0.003). See Table 3.

### 3.4. Survival Analysis after SPC

Figure 2 shows the overall survival after SPC stratified by HPV status. Median follow-up time after SPC was 1.1 years (IQR = 0.46–2.27 years). One-year overall survival for the HPV- OPSCC group was 63.3% [95%CI: 56.3–71.9%], and the three- and five-year overall survivals were 31.5% [95%CI: 23.0–43.3%] and 24.2% [95%CI: 15.6–37.6%], respectively. For HPV+ OPSCC patients, one-year survival was 72.1% [95%CI: 64.8–80.2%], three-year survival was 50.2% [95%CI: 41.6–60.7%], and the five-year survival was 43.1% [95%CI: 33.7–55.1%]. There was a significant difference between the two groups (*p* = 0.003), as patients with HPV+ OPSCC had a better survival.

Figure 3 shows the overall survival after SPC stratified by SPC localized “within” or “outside” LHN. Patients with SPC localized outside the LHN had a significant better survival (*p* = 0.01), and the one-, three-, and five-year overall survival was 70.6% [95%CI: 63.7–78.4%], 52.3% [95%CI: 43.9–62.2%], and 45.1% [95%CI: 35.8–57.0%]. For patients with SPC within LHN, the one-, three-, and five-year overall survival was 64.3% [95%CI: 56.6–73.1%], 28.6% [95%CI: 20.3–40.4%], and 20.8% [95%CI: 12.6–34.3%], respectively.

## 4. Discussion

To our knowledge, this is the largest consecutive, population-based retrospective study including 2584 patients diagnosed with OPSCC that all were HPV- and p16-tested. In this cohort from eastern Denmark from years 2000–2020, we found a high risk of SPC following a primary OPSCC. The risk was nearly five times higher (IRR = 4.96) compared to the risk of any cancer (the same type of malignancies) in the general population in Denmark [28]. It is expected that head and neck cancer patients are more likely to develop SPCs since they smoke and drink more compared to the general population [30]. This applies especially to patients diagnosed with SPC in the upper aerodigestive tract, the lungs, or the oesophagus, where the exposure of these carcinogens is most concentrated. Our study supports this theory since most patients with SPC were former or current smokers, and significantly more patients with SPC had a smoking history compared to patients without SPC. León, X. et al. previously found an increased risk of SPC for patients continuing tobacco smoking and alcohol consumption after treatment of a head and neck cancer, including OPSCC [31]. Whether patients continued to smoke tobacco and consume alcohol or had quit after their primary OPSCC diagnosis was unfortunately not reported in this study, but the previous finding by León, X. et al. [31] should be used in guidance of patients to improve prognosis after an OPSCC diagnosis.

We found an increased risk of SPC in older patients, which is not surprising since age is a well-established risk factor in all types of cancer. We also saw a higher risk of SPC in patients with a UICC8 stage II OPSCC at the time of diagnosis, which calls for further investigation since we cannot describe a clear explanation. Patients with a N2 classification at the time of diagnosis of OPSCC were shown to have a significantly lower risk of SPC in our analysis. This might be because HPV+ patients more often have lymph node involvement at the time of diagnosis. It would be of interest to examine baseline characteristics stratified by HPV status to further investigate the observed association. The risk of SPC was not associated with T-stage OPSCC or sublocation. HPV-positivity showed to be a significant protecting factor, which is in line with the before-mentioned differences in baseline characteristics between the subgroups of HPV+ and HPV- OPSCC patients. Further, Bosshart, S.L. et al. [32] also found HPV+ to be a significant protecting factor for SPC, which substantiates our finding. However, they did not find an association between either age or tumour stage and the risk of SPC. The last-mentioned inconsistency and the association between tumour stage and the risk of SPC following an OPSCC calls for further research. 

Most patients with OPSCC received primary radiation therapy with or without chemotherapy, which is a well-known risk factor for developing SPC, but according to Hashibe, M. et al. [33], the earliest radiation-induced SPCs for solid tumours are detected 10 years after treatment. Since we only had four patients diagnosed with SPC in the head and neck region later than ten years after the index tumour, we do not consider radiation as the major cause of SPC in this study.

The overall predominant sites of SPC were the lungs, head, and neck, and the gastrointestinal tract. The SPC localizations differed between patients with HPV+ and HPV- OPSCC index tumours and head and neck were only the third most frequent site for HPV+ OPSCC compared to the second most frequent site in HPV- OPSCC. In contrast to patients with HPV- OPSCC, gender-specific SPCs were more common for patients with HPV+ OPSCC than head and neck malignancies. Further, HPV+ patients had a better overall survival than HPV- patients, reflecting the well-established clinical differences between patients with HPV+ and HPV- OPSCC. Since the lungs are the most predominant site for both HPV+ and HPV- and since the lungs are a common site for distant metastasis from OPSCC, we find it relevant, in the future, to further studies to investigate new malignancies in the lungs following OPSCC to better differentiate between SPC and recurrence [34]. Further, the different distribution of SPC localizations could be linked to an immunosuppressed state, which was described in patients with HPV+ OPSCC earlier [35,36]. Interestingly, the incidence of SPC within the LHN was significantly different in HPV+ and HPV- OPSCC (*p* = 0.003), at 37.6% and 55.0%, respectively. We also found that patients with SPC outside the LHN area had a significant and better survival compared to patients with SPC within the LHN (*p* = 0.01), which probably reflects the inferior survival of lung cancer and the challenges of treating a second cancer in the head and neck area. Holstead, R. et al. [23] also found an association between HPV+ OPSCC and SPC outside the LHN area and requested further investigation for development of individualized guidelines. Due to the high risk of SPC following OPSCC, clinicians should be aware of this in the post-treatment surveillance of patients, especially in the first years after diagnosis (median time to SPC 2.0 years). The attention should be further increased in HPV- patients since they develop SPC significantly earlier than HPV+ OPSCC patients (*p* = 0.001) and have a worse survival, which is in line with other studies [21,24]. Additional, especially the LHN area should be investigated close for early detection of SPC to improve survival.

According to HPV status and the overall risk of SPC, we found a significantly lower risk of SPC in patients with HPV+ OPSCC index tumours. Other studies also found a lower risk of SPC in patients with p16+ OPSCC [22,32] but did not use double positivity for both HPV DNA and p16, as in this study. Based on the same database, Grønhøj, C. et al. [34] established earlier that HPV+ patients are less likely to have both distant- (*p* = 0.02) and loco-regional (*p* = 0.001) recurrence following OPSCC. In this study, we found a significantly better overall survival following SPC for HPV+ OPSCC compared to HPV- (*p* = 0.003). These findings could be due to the patients diagnosed with HPV+ OPSCC being younger, demonstrating less comorbidity, and having a more favourable smoking history, as shown earlier [37]. Furthermore, this might also reflect the differences in SPC localization in HPV+ OPSCC and HPV- OPSCC patients, with the incidence of SPC within the LHN area being significantly lower in the HPV+ OPSCC group. Most patients in this study were treated with radiotherapy or chemoradiotherapy, which challenges treatment of a SPC in the head and neck area since a second radiation in the same area is often not an option. This might explain the inferior survival in patients with SPC in the head and neck area, which is shown to be more represented in HPV- patients and might support their poorer survival.

The current study was restricted by its retrospective design. Further, there is no international consensus on how to define an SPC, and hence, the definition differs between studies [3,20,21,31,38]. In this study, we considered all second tumours within five years after primary OPSCC diagnosis at the same localization as a recurrence of the index tumour though it could be a potential SPC. This might affect the total number of SPC. A worldwide restricted consensus on the definition of SPC could be useful when comparing studies. 

The strength of this study was the large size of the population-based, non-selected, consecutive cohort of all patients diagnosed with OPSCC in eastern Denmark with a high number of events, appropriate follow-up time, and a broad knowledge of clinical data and tumour characteristics, including both HPV DNA and p16 status. Further, it was a strength that all patients received a standardized treatment through the whole period, following the same guidelines.

## 5. Conclusions

This study identified a high incidence of SPC (12.3%) among 2584 patients following a primary OPSCC in eastern Denmark from 2000–2020. The risk of SPC in patients diagnosed with OPSCC was nearly five times higher (IRR = 4.96) compared to the risk of cancer in the general population in Denmark. The most frequent localization of SPC was the LHN, but a different distribution was found in HPV+ and HPV- OPSCC, with SPC within the LHN being significantly lower in HPV+ OPSCC. The survival after SPC was significant better in HPV+ OPSCC and patients with SPC outside the LHN area.

This knowledge should be considered during post-treatment surveillance and might guide targeted imaging for early detection and thereby improve prognostication.

## Figures and Tables

**Figure 1 viruses-15-00034-f001:**
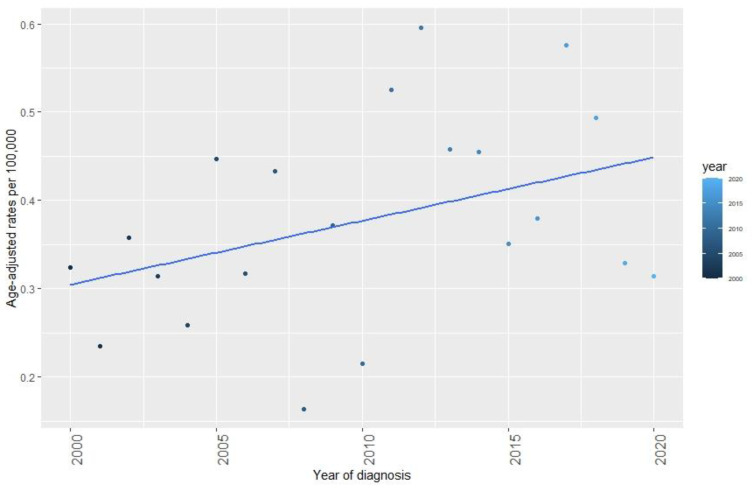
Age-adjusted incidence rates (AAIR) per 100,000 of SPC following treatment of OPSCC in Eastern Denmark from 2000–2020. The total incidence of SPC showed an overall increase through the period 2000 to 2020.

**Figure 2 viruses-15-00034-f002:**
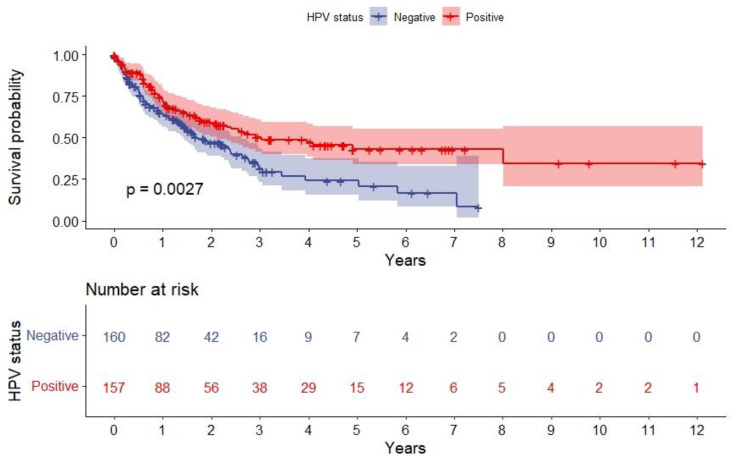
Kaplan–Meier plot depicting overall survival after SPC stratified on HPV status. The red line shows patients with HPV+ OPSCC, and the blue line shows patients with HPV- OPSCC. The difference is significant (*p* = 0.003), tested with the log-rank method. Abbreviation: HPV, human papillomavirus; SPC, second primary malignancy.

**Figure 3 viruses-15-00034-f003:**
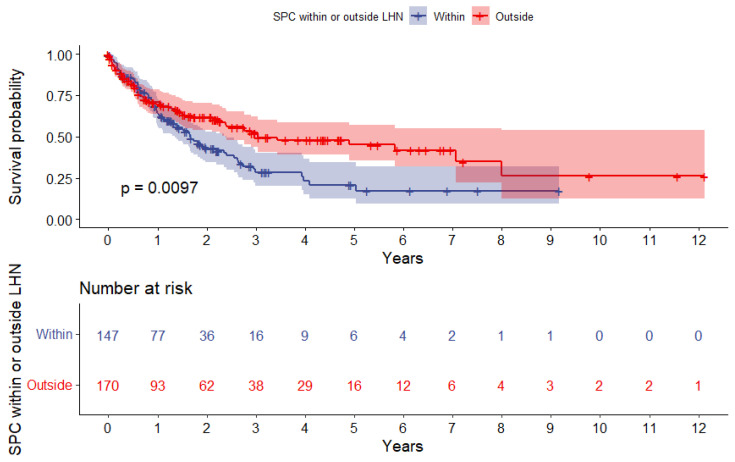
Kaplan–Meier plot depicting overall survival after SPC stratified by SPC localized within or outside the LHN. The red line shows patients with SPC outside LHN, and the blue line shows patients with SPC within LHN. The difference is significant (*p* = 0.01) tested with the log-rank method. Abbreviation: SPC, second primary malignancy; LHN, lungs, head, and neck.

**Table 1 viruses-15-00034-t001:** Baseline characteristics of patients with and without second primary cancer (SPC) following primary oropharyngeal squamous cell carcinoma (OPSCC) diagnosed in eastern Denmark during the period 2000–2020.

	Total		SPC- (*n* = 2267, 87.7%)		SPC+ (*n* = 317, 12.3%)		*p*-Value *
**Median age at diagnosis, (range) years**	61	(30–92)	60	(30–92)	64	(37–86)	<0.001 *
**Sex, *n* (%)** Female Male	2584 697 1887	(100%) (27.0%) (73.0%)	609 1658	(26.9%) (73.1%)	88 229	(27.8%) (72.2%)	0.736
**UICC8 T-classification, *n* (%)** 1 2 3 4	**2575**^a^ 635 1065 529 346	**(100%)** (24.7%) (41.4%) (20.5%) (13.4%)	559 926 468 306	(24.7%) (41.0%) (20.7%) (13.5%)	76 139 61 40	(24.1%) (44.0%) (19.3%) (12.7%)	0.896
**UICC8 N-classification, *n* (%)** 0 1 2 3	**2584** 589 1306 580 109	**(100%)** (22.8%) (50.5%) (22.4%) (4.2%)	488 1161 522 96	(21.5%) (51.2%) (23.0%) (4.2%)	101 145 58 13	(31.9%) (45.7%) (18.3%) (4.1%)	<0.001 *
**UICC8 M-classification, n (%)** 0 1	**2569**^a^ 2551 18	**(100%)** (99.3%) (0.7%)	2237 16	(99.3%) (0.7%)	314 2	(99.4%) (0.6%)	1.000
**UICC8 Stage, n (%)** I II III IV	**2518**^a^ 1127 489 427 475	**(100%)** (44.8%) (19.4%) (17.0%) (18.9%)	1006 414 371 415	(45.6%) (18.8%) (16.8%) (18.8%)	121 75 56 60	(38.8%) (24.0%) (17.9%) (19.2%)	0.077
**Alcohol consumption **, n (%)** No abuse Previous or current abuse	**2583**^a^ 1059 1524	**(100%)** (41.0%) (59.0%)	951 1315	(42.0%) (58.0%)	108 209	(34.1%) (65.9%)	0.007 *
**Smoking status, n (%)** Never smoker Former smoker Current smoker	**2555**^a^ 566 1001 988	**(100%)** (22.2%) (39.2%) (38.7%)	513 862 867	(22.9%) (38.4%) (38.7%)	53 139 121	(16.9%) (44.4%) (38.7%)	0.032 *
**Histopathological type, *n* (%)** Non-keratinizing Keratinizing/hybrid/other	**2514**^a^ 1378 1136	**(100%)** (54.8%) (45.2%)	1240 966	(56.2%) (43.8%)	138 170	(44.8%) (55.2%)	0.002 *
**HPV status (HPV DNA/p16), *n* (%)** HPV-negative HPV-positive	**2584** 1052 1532	**(100%)** (40.7%) (59.3%)	892 1375	(39.3%) (60.7%)	160 157	(50.5%) (49.5%)	<0.001 *
**Primary tumour location, *n* (%)** Palatine tonsils Base of tongue Other	**2584** 1308 762 514	**(100%)** (50.6%) (29.5%) (19.9%)	1149 688 430	(50.7%) (30.3%) (19.0%)	159 74 84	(50.2%) (23.3%) (26.5%)	0.002 *
**Performance score, *n* (%)** 0 1–4	**2248**^a^ 1697 551	**(100%)** (75.5%) (24.5%)	1499 475	(75.9%) (24.1%)	198 76	(72.3%) (27.7%)	0.211
**Prior cancer, *n* (%)** No Yes	**2575**^a ^ 2308 267	**(100%)** (89.6%) (10.4%)	2035 226	(90.0%) (10.0%)	273 41	(86.9%) (13.1%)	0.095
**Chemotherapy type, *n* (%)** Cisplatin Other ***	**1194**^b^ 1094 100	**(100%)** (91.6%) (8.4%)	986 94	(91.3%) (8.7%)	108 6	(94.7%) (5.3%)	0.207

Factors significantly different in patients with and without SPC were age, UICC8 N-stage, alcohol consumption, smoking status, carcinoma type, HPV status, and primary tumour localization.Abbreviations: SPC, second primary malignancy; SPC-, patients without a SPC; SPC+, patients with a SPC; HPV, human papillomavirus; UICC8, Union for International Cancer Control eighth edition; HPV status (HPV DNA and p16), only double positive counts for HPV-positive. * Statistical significance. The Pearson’s chi^2^ test was used for statistical significance for the categorical variables. *t*-test was used for significance on the age variable and Fisher’s test for UICC8 M-classification. ** Female: alcohol abuse is defined as >7 units per week. Male: alcohol abuse is defined as >14 units per week. *** Carboplatin, epidermal growth factor receptor (EGFR) inhibitors, cisplatin + EGFR inhibitors, or “other”. ^a^ The total number of cases differs due to unavailable data per variable. **^b^** Total number of patients who received chemotherapy.

**Table 2 viruses-15-00034-t002:** Univariable and multivariable analysis of possible risk factors for SPC.

Risk Factors	Univariable HR [95% CI]	*p*-Value	Multivariable HR [95% CI]	*p*-Value
**Age**	1.05 [1.04–1.06]	<0.001 *	1.04 [1.03–1.06]	<0.001 *
**Sex** Female Male	Ref. 1.03 [0.80–1.31]	- 0.836	Ref. 1.15 [0.89–1.48]	- 0.287
**Sublocation** Palatine tonsils Base of tongue Other	Ref. 0.91 [0.69–1.21] 1.82 [1.39–2.37]	- 0.524 <0.001 *	Ref. 0.80 [0.61–1.07] 1.02 [0.75–1.39]	- 0.132 0.887
**UICC8-stage** I II III IV	Ref. 1.76 [1.32–2.35] 1.84 [1.34–2.53] 2.47 [1.80–3.39]	- <0.001 * <0.001 * <0.001 *	Ref. 1.80 [1.20–2.69] 1.60 [0.99–2.59] 1.94 [0.98–3.87]	- 0.004 * 0.057 0.058
**UICC8 T-classification** T1 T2 T3 T4	Ref. 1.02 [0.77–1.35] 1.19 [0.85–1.66] 1.60 [1.09–2.35]	- 0.900 0.323 0.017 *	Ref. 0.86 [0.64–1.15] 0.67 [0.44–1.04] 0.91 [0.53–1.54]	- 0.302 0.072 0.722
**UICC8 N-classification** N0 N1 N2 N3	Ref. 0.52 [0.40–0.66] 0.74 [0.53–1.02] 0.81 [0.45–1.44]	- <0.001 * 0.068 0.465	Ref. 0.80 [0.60–1.07] 0.61 [0.38–0.97] 0.77 [0.37–1.57]	- 0.129 0.036 * 0.466
**Smoking status** Never smoker Former smoker Current smoker	Ref. 1.63 [1.19–2.24] 1.89 [1.37–2.62]	- 0.002 * <0.001 *	Ref. 1.03 [0.94–1.80] 1.10 [0.77–1.59]	- 0.112 0.594
**HPV status (HPV DNA/p16)** HPV-negative HPV-positive	Ref. 0.411 [0.33–0.52]	- <0.001 *	Ref. 0.56 [0.40–0.80]	- 0.001 *

Table 2 shows results of the univariable and multivariable Cox regression analysis of SPC on the independent variables of age, sex, tumour sublocation, UICC8-stage, UICC8 T-classification, UICC8 N-classification, smoking status, and tumour HPV status (DNA and p16), only double positive counts for HPV-positive. Abbreviations: HPV, human papillomavirus; HR, hazard ratio; CI. confidence interval; UICC8, Union for International Cancer Control eighth edition; SPC, second primary cancer. * Statistical significance.

**Table 3 viruses-15-00034-t003:** Localizations of SPC following OPSCC in eastern Denmark, overall and stratified by HPV status.

			HPV-		HPV+		*p*-Value
**Second primary cancer localization, *n* (%)** Lung Head and neck Gastrointestinal tract Skin ** Gender-specific *** Haematologic cancer Breast Liver or pancreas Urologic cancer CNS or brain Non-oropharyngeal HPV- Associated **** Other	**317** 95 52 47 29 24 21 18 11 11 4 3 2	**(100%)** (30.0%) (16.4%) (14.8%) (9.1%) (7.6%) (6.6%) (5.7%) (3.5%) (3.5%) (1.3%) (0.9%) (0.6%)	**160** 57 31 29 12 1 5 10 6 4 1 2 2	**(50.5%)** (35.6%) (19.4%) (18.1%) (7.5%) (0.6%) (3.1%) (6.2%) (3.8%) (2.5%) (0.6%) (1.3%) (1.3%)	**157** 38 21 18 17 23 16 8 5 7 3 1 0	**(49.5%)** (24.2%) (13.4%) (11.5%) (10.8%) (14.6%) (10.2%) (5.1%) (3.2%) (4.5%) (1.9%) (0.6%) (0.0%)	
**Localizations stratified by LHN, *n* (%)** Within LHN Outside LHN			88 72	(55.0%) (45.0%)	59 98	(37.6%) (62.4%)	0.003 *

Table 3 shows the number of SPC distributed on tumour localization in total and stratified on HPV status. * There was a significant difference between the two groups (*p* = 0.003) using the Pearson’s chi^2^ test. ** Squamous cell carcinoma or melanoma. *** Prostate, ovaries, or endometrium. **** Anal, vulvar, vaginal, or penile. Abbreviation: HPV, human papillomavirus; SPC, second primary malignancy; OPSCC, oropharynx squamous cell carcinoma; LHN, lungs, head, and neck.

## Data Availability

Publicly archived datasets analysed or generated during the study are shown in the following references [28,29,30]. See Data Availability Statements in section “MDPI Research Data Policies” at https://www.mdpi.com/ethics, accessed on 1 December 2022.

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
