# Peer review of "The Incidence, Survival, and HPV Impact of Second Primary Cancer following Primary Oropharyngeal Squamous Cell Carcinoma: A 20-Year Retrospective and Population-Based Study"

_viruses, 2022, doi:10.3390/v15010034_

Round 1

Reviewer 1 Report

Dear Authors,

You present here a retrospective study regarding the incidence, survival and HPV impact of second cancer following primary oropharingeal cancer. The paper is well written and organized, with just minor English mistakes. 

I suggest you add the drugs used in chemotherapy and if there were differences  regarding the development of second cancer in relationship with the chemotherapy used.

Plus, I suggest you investigate more the data regarding the survival of the patients and what were the medical protocols used (chemotherapy, radiotherapy). 

Reviewer 2 Report

The paper idea is novel and contributes well to the scientific community. I congratulate the authors on producing an extensive master piece of big data over 20 years.

The statistical powers used are great but the results must be clearly discussed. There are a lot of graphs and tables and the results are only presented the same way as in the Tables and Figures. The statistical inference should be properly explained.
